# Anapole mediated giant photothermal nonlinearity in nanostructured silicon

Tianyue Zhang[1,7], Ying Che[1,2,7], Kai Chen [1], Jian Xu[1], Yi Xu [3], Te Wen[4], Guowei Lu [4], Xiaowei Liu[1], Bin Wang[2], Xiaoxuan Xu[2], Yi-Shiou Duh[5], Yu-Lung Tang[5], Jing Han[1], Yaoyu Cao[1], Bai-Ou Guan[1], Shi-Wei Chu [5,6✉] & Xiangping Li [1✉]

Featured with a plethora of electric and magnetic Mie resonances, high index dielectric nanostructures offer a versatile platform to concentrate light-matter interactions at the nanoscale. By integrating unique features of far-field scattering control and near-field concentration from radiationless anapole states, here, we demonstrate a giant photothermal nonlinearity in single subwavelength-sized silicon nanodisks. The nanoscale energy concentration and consequent near-field enhancements mediated by the anapole mode yield a reversible nonlinear scattering with a large modulation depth and a broad dynamic range, unveiling a record-high nonlinear index change up to 0.5 at mild incident light intensities on the order of MW/cm². The observed photothermal nonlinearity showcases three orders of magnitude enhancement compared with that of unstructured bulk silicon, as well as nearly one order of magnitude higher than that through the radiative electric dipolar mode. Such nonlinear scattering can empower distinctive point spread functions in confocal reflectance imaging, offering the potential for far-field localization of nanostructured Si with an accuracy approaching 40 nm. Our findings shed new light on active silicon photonics based on optical anapoles.

[1] Guangdong Provincial Key Laboratory of Optical Fiber Sensing and Communications, Institute of Photonics Technology, Jinan University, 510632 Guangzhou, China. [2] The Key Laboratory of Weak-Light Nonlinear Photonics, Ministry of Education, School of Physics, Nankai University, 300071 Tianjin, China. [3] Department of Electronic Engineering, College of Information Science and Technology, Jinan University, 510632 Guangzhou, China. [4] State Key Laboratory for Mesoscopic Physics, Frontiers Science Center for Nano-optoelectronics & Collaborative Innovation Center of Quantum Matter, School of Physics, Peking University, 100871 Beijing, China. [5] Department of Physics, National Taiwan University, No. 1, Sec. 4, Roosevelt Rd., 10617 Taipei, Taiwan. [6] Brain Research Center, National Tsing Hua University, 101, Sec 2, Guangfu Road, 30013 Hsinchu, Taiwan. [7] These authors contributed equally: Tianyue Zhang, Ying Che. ✉email: swchu@phys.ntu.edu.tw; xiangpingli@jnu.edu.cn

All-dielectric, high refractive index nanostructures offer unique ability to efficiently confine and manipulate light at the nanoscale based on their potentials to control both optically induced electric and magnetic Mie resonances[1–4]. During recent years, the interplays of a wealth of Mie-type resonant modes have unveiled many novel physical phenomena, such as unidirectional scattering[5–8], magnetic Fano resonances[9], bound states in the continuum[10,11], and nonradiating optical anapoles[12,13]. Among these observations, which originate from multimodal interference in dielectric nanostructures, optical anapole holds distinct features characterized by vanishing far-field scattering accompanied with strong near-field absorptions. The former is a result of far-field destructive interference between a toroidal dipole (TD) and an out-of-phase oscillating electric dipole (ED)[14], and the latter is due to the induced displacement currents inside the nanostructures, which produce tightly confined near-fields to resonantly enhance the local density of photonic states.

The discovery of the general existence of optical anapoles in dielectric nanostructures immediately spurred extensive investigations on diverse applications. Engineered anapole states have been used to tailor light scattering in the far-field for inducing optical transparency[15,16] or rendering pure magnetic dipole source[17]. Functional anapole metamaterials and metasurfaces featuring high-quality factors have revealed potentials in optical modulation and sensing[18,19]. More importantly, energy concentration in the subwavelength volume associated with anapole-mediated hotspots facilitates boosting near-field light–matter interactions including nonlinear harmonic generation[20–22], nanoscale lasing[23], broadband absorption[24], strong coupling with plasmon[25] or molecular excitons[26,27], and enhanced Raman spectroscopy[28,29]. To further advance nanophotonic devices, the full potential amalgamating the benefits in both far-field and near-field features from optical anapoles remains tantalizing.

In this report, we discover a giant photothermal nonlinearity mediated by anapole states within a subwavelength-sized silicon (Si) nanodisk and demonstrate dynamic scattering modulations.

Leveraging their nontrivial electromagnetic near-fields, anapole modes boost photothermal nonlinearity by three orders of magnitude higher than that of bulk Si, or nearly one-order-of-magnitude outperforming the radiative ED-driven enhancement for similar sized Si nanostructures. A record-high photothermal refractive index change $\Delta n$ up to 0.5 can be achieved upon a mild laser radiance intensity of 1.25 MW/cm² through optically pumping a Si nanodisk at a wavelength close to the anapole mode. The giant photothermal nonlinearity thus offers an active mechanism for dynamic tuning of far-field radiation from multipolar modes. The scattering cross section that is normalized to 1 for linear scattering can be reversibly suppressed down to 0.1 and then rapidly enhanced up to 1.1, demonstrating a large modulation depth and a broad dynamic range, due to the progressive transition of dominant modes from the bright state to the low-radiating dark state and further moving towards the bright state again. Consequently, we found distinctive point spread functions (PSFs) in confocal reflection imaging induced by the nonlinear scattering modulation of single Si nanodisks. These PSFs can be employed for optical localization of Si nanostructures in dense arrays with an accuracy approaching 40 nm. Compared to existing techniques, anapole-mediated photothermal nonlinearity offers noninvasive all-optical modulation of scattering, shedding new light on active photonics harnessing dielectric nanostructures for on-demand tunability.

## Results

**Nonlinear scattering of Si nanodisks.** The principle is schematically illustrated in Fig. 1a. A Si nanodisk illuminated by a continuous-wave (CW) laser beam converts incident light into heat, which raises the disk temperature substantially. Photothermal mechanism induces the refractive index variation, resulting in continuous red-shifting of Mie resonances under light excitation, and thus allows for tunable optical responses of the silicon nanodisk. With rationally designed dimensions of the Si nanodisk, the associated anapole mode can be actively engineered

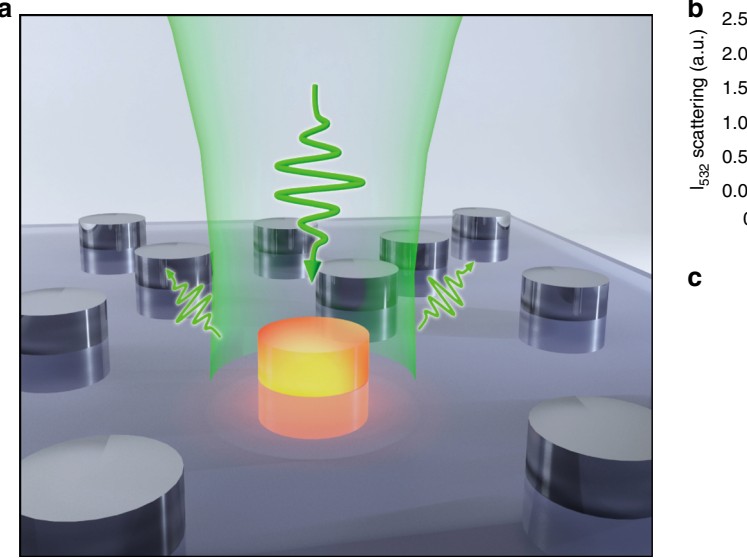

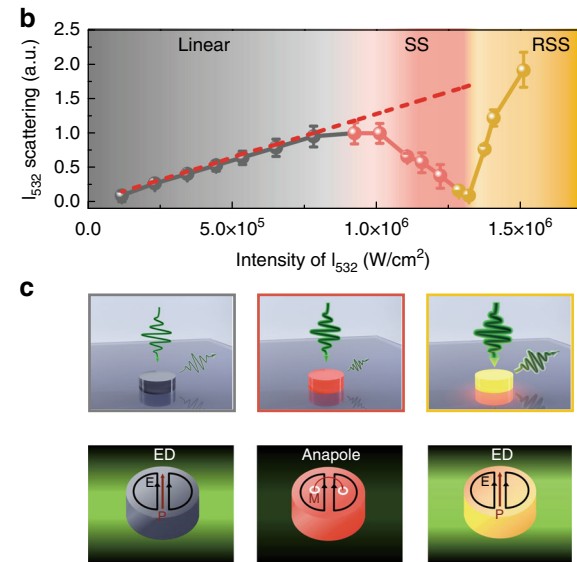

**Fig. 1 Schematic illustration and experimental observation of anapole-mediated photothermal nonlinearity. a** Illustration of strong optical heating that efficiently converts light into temperature rises within subwavelength volume of Si nanodisks. **b** The nonlinear dependency of scattering on irradiance intensities in single Si nanodisks for excitations at the wavelength of 532 nm. When the excitation intensity is low, the scattering is linearly proportional to excitation irradiances (denoted as the dash line). When the excitation intensity exceeds $8 \times 10^5$ W/cm², scattering deviates from the linear trend into deep saturation. When the excitation intensity is higher than $1.3 \times 10^6$ W/cm², the scattering sharply increases, showing reverse saturation scattering. The error bars represent the standard deviations of scattering intensities based on statistics of twelve nanodisks. **c** Schemes of anapole-driven nonlinear scattering due to progressive transition of dominant modes from the bright state to the radiationless anapole state and further moving towards the bright state again.

and tuned in the vicinity of the excitation wavelength. Hence, the illumination irradiance required for the large refractive index modification of the Si nanodisk can be efficiently reduced via anapole-assisted absorption enhancement. As the laser intensity increases, the dominant mode at the excitation wavelength transits from the initial bright mode towards the anapole mode (will be discussed in detail below), which induces saturation scattering (SS)[30] and significant reduction of the scattering intensity (Fig. 1b, c). Further temperature rise leads to the progressive transition from the nonradiating anapole to the ED mode, inducing a sharp increase of the scattering intensity, which we denote as reverse saturation scattering (RSS). We will show in the following paragraphs that anapole-driven nonlinear scattering herein can be actively controlled in a reversible manner without the need to physically alter the dimensions of the nanostructure or change the environment.

The anapole state is an engineered destructive interference between toroidal and electric dipoles presenting in well-designed dielectric nanostructures. Pioneering works have uncovered that disk geometry, with its structural simplicity, supports the fundamental and higher-order anapole modes[14,31]. To generate anapole mode at the vicinity of the excitation wavelength of 532 nm, Si nanodisks with diameter D of 200 nm and height h of 50 nm were used in the present study. Well-dispersed silicon nanodisks on a glass substrate were fabricated by colloidal-mask lithography[32,33] with step-by-step fabrication sketched in Fig. 2a (see Methods). The as-prepared Si nanodisks were laser annealed before the nonlinear scattering study (Supplementary Note 1 and Supplementary Fig. 1). Scanning electron microscopy (SEM) images (Fig. 2b) show high-quality Si nanodisks with an average spacing of micrometers to avoid the coupling effect. Scattering images of individual Si nanodisks under the 532 nm CW laser illumination are measured with a reflectance confocal laser scanning microscope (Fig. 2c, see Methods).

We examine the photothermal nonlinearity through analyzing the scattering PSFs of single nanodisks[30,34–37]. Figure 2d, e depicts the evolution of the PSFs from isolated Si nanodisks by increasing excitation intensities. When the laser intensity is low, the PSF fits well to a Gaussian function with a full width at half maximum (FWHM) of 230 nm. The shape of the PSF changes dramatically when SS occurs. At deep saturation, a doughnut-shaped PSF appears with a low intensity in the center (Fig. 2d (d-3)), representing the radiationless anapole state. By further increasing the excitation intensity, a sharp peak emerges from the doughnut center, indicating the onset of RSS (shown in Fig. 2d (d-4)). The strong central peak starts to dominate the PSF as the irradiance intensity continues to increase (Fig. 2d (d-5)).

The evolution of PSF profiles can be numerically reproduced through modelling confocal reflectance imaging of a subwavelength object displaying intensity-dependent nonlinear scattering (Supplementary Note 2 and Supplementary Fig. 2). Such nonlinear behavior is quantified by taking the ratio of scattering over excitation intensity extracted from experimental results in Fig. 1b. As shown in Fig. 2f, the normalized scattering cross section stays constant (normalized to 1) at the initial linear region, then decreases to 0.1 for the largest SS, and again drastically rises to 1.1 for RSS, demonstrating a large dynamic range spanning from scattering suppression to enhancement. Throughout the nonlinear scattering measurements, the full recovery of both scattering intensities and corresponding PSFs confirms its reversibility (Fig. 2g). The reversibility is also checked by the reversible evolution of normalized scattering cross sections with excitation intensities varied between low and high (Supplementary Fig. 3). This ensures the scattering behavior of a single Si nanodisk can be actively and reversibly engineered.

**Anapole-mediated photothermal nonlinearity.** The anapole mode supported by the Si nanodisk is verified by both simulation and experimental measurements (Supplementary Fig. 4). The anapole state is featured by a significant dip in the total far-field scattering spectrum, accompanied by unique near-field distributions as shown in Fig. 3a, b. Notably, the boosted near-field energy directly contributes to the absorption peak at the anapole wavelength, leading to a substantial temperature rise within the Si nanodisk. To corroborate the local temperature rise, we perform Raman spectroscopy measurements at different irradiance intensities at the wavelength of 532 nm. By taking the intensity ratio of anti-Stokes to Stokes Raman spectra[38], temperature increment within the Si nanodisks under various incident intensities can be extracted (see Methods and Supplementary Fig. 5). Experimental results from Raman thermometry reveal that the Si nanodisks experience a huge temperature rise, more than 900 °C above room temperature (RT) during the nonlinear scattering processes (Fig. 3c). Such substantial temperature rises of Si nanostructures, particularly in the thermally sensitive visible wavelength region[39,40], induces large modifications of refractive indices in both real and imaginary parts, also known as thermo-optic effect[41,42] (Supplementary Fig. 6). In the temperature range from RT to 950 °C, the change of the refractive index in real part $\Delta n$ is extrapolated to be 0.5 at a moderate laser intensity of $1.25 \times 10^6$ W/cm². This equivalently gives the effective nonlinear refractive index as $n_{2,@532\,nm} = \Delta n/I = 0.4$ cm²/MW. Compared with the measured temperature rise in bulk Si, i.e., less than 10 °C under much higher laser intensities, optical anapole significantly enhances photothermal nonlinearity by three orders of magnitude (Supplementary Fig. 7). Since the absorption of the nanodisk depends on the contribution from all multipole modes, it keeps increasing with temperature (Fig. 3d). In contrast, the scattering can be desirably manipulated in response to optical heating of the anapole mode, thus yielding unconventional nonlinear scattering responses.

The simulated backward scattering cross-section $C_{scaB}$ is plotted in Fig. 3e to show the spectral response and the tuning range of Mie resonances with temperature increments. The pronounced scattering maxima and minima (marked by the dashed lines for eye guidance) undergo continuous redshifts with elevated temperatures. The corresponding photothermally-induced Mie resonance shifts are estimated to be $\Delta\lambda \approx \lambda\Delta n/n \approx 50$ nm, or $\Delta\lambda/\lambda \sim 10\%$, which is indeed observed in Fig. 3e. The large resonance tuning represents a significant improvement compared with previous studies using liquid crystals or thermo-optic effects[43–46]. We also remark here that free-carrier contributions are ruled out by the fact that they lead to a negative $\Delta n$, which causes the blue shift of the resonance[47].

The irradiance-induced temperature rise from RT to 500–600 °C allows for suppressing the backward scattering cross section from $2.3 \times 10^{-14}$ m² to $0.6 \times 10^{-14}$ m² (Fig. 3e–g), corresponding to 74% modulation. This agrees qualitatively with the experimental observation of 90% suppression of normalized cross section from linear to SS in Fig. 2f. The large modulation depth is attributed to the fact that the excitation laser delicately operates in the vicinity of the anapole mode. The sharp slope in the far-field scattering spectrum near the anapole mode enables pronounced changes of backward scattering cross sections via a small spectral tuning (Fig. 3f). The bottom panels in Fig. 3f further depict the near-field distributions excited at the wavelength of 532 nm, providing a clear progressive transition that the illumination laser initially excites the lower-energy-side of the anapole mode, and then gradually approaches resonant with the anapole mode and finally excites the ED mode. To establish the relationship between scattering $I_{sca}$ and the incident intensity $I_{exc}$ for single Si

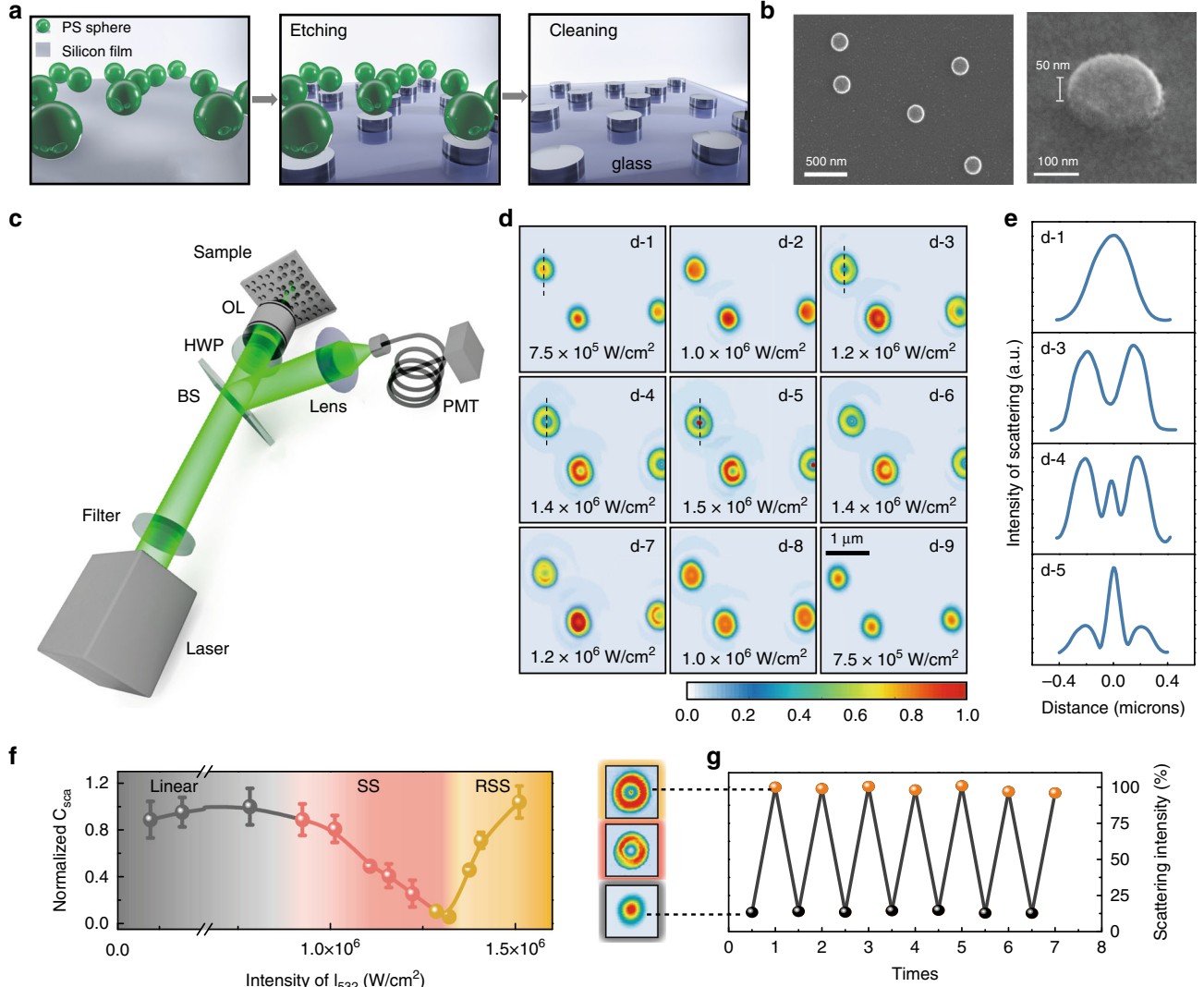

**Fig. 2 Nonlinear scattering measurements. a** Schematic fabrication processes of isolated Si nanodisks. **b** SEM images as well as 30° tilted view, showing the resulting Si nanodisks with diameter of 200 nm and height of 50 nm. **c** Optical setup of the reflected laser scanning confocal microscope. HWP half-wave plate, BS beam splitter, OL objective lens, PMT photomultiplier tube. **d** Measured PSFs under different laser intensities at the wavelength of 532 nm. **e** The intensity lateral profile of the selected nanodisks (black dashed lines in **d**). **f** The evolution of normalized scattering cross section with excitation intensities. The error bars show the standard deviations of normalized scattering cross sections according to statistics of 12 nanodisks. **g** Reversibility of nonlinear scattering is confirmed by the full recovery of scattering intensities as well as corresponding PSFs from the same nanodisk under repetitive measurements.

nanodisks, the intensity-dependent nonlinear scattering can be derived as $I_{sca} \propto C_{sca} \cdot I_{exc}$, which is depicted in Fig. 3h. It shows a similar trend with the experimental results in Fig. 1b. Although we recorded only the backward scattering in experiments, the temperature dependences of simulated total scattering and forward scattering reveal a similar trend (Supplementary Fig. 8), thus excluding the scattering modulation originating from energy redistributions between forward and backward radiation[43].

To illustrate the important role played by anapole modes, we performed calculations for another two representative sizes of Si nanodisks (Supplementary Note 9). For a smaller-sized nanodisk (D = 170 nm), the overall photothermal tuning occurs near its ED mode. We show that ED-mediated process presents much weaker photothermal nonlinearity by a moderate temperature rise less than 200 °C at a similar laser intensity of $1.25 \times 10^{6}$ W/cm². The corresponding nonlinear refractive index $n_2 = 0.08$ cm²/MW, which is five times lower than anapole-assisted process. In

addition, its scattering cross sections keep almost unchanged within the excitation intensity range, resulting in negligible SS. On the contrary, for a larger-sized nanodisk (D = 230 nm), its original anapole mode coincides with the excitation wavelength at RT. Elevated temperatures induce redshifts of the anapole mode away from the excitation wavelength, leading to a monotonical increase of scattering cross sections. Thus, a sharp RSS is achieved. Similar nonlinear scattering has been reported by optically heating the magnetic quadrupole[48].

**Potential for far-field optical localization of Si nanostructures.** Leveraging the photothermal nonlinear scattering, we demonstrate the potential for far-field optical localization of Si nanostructures. In analogy to differential excitation methods[49–51], the difference between the two scattering images obtained at RSS and SS stages yields a narrow spot with a sub-diffractive FWHM as

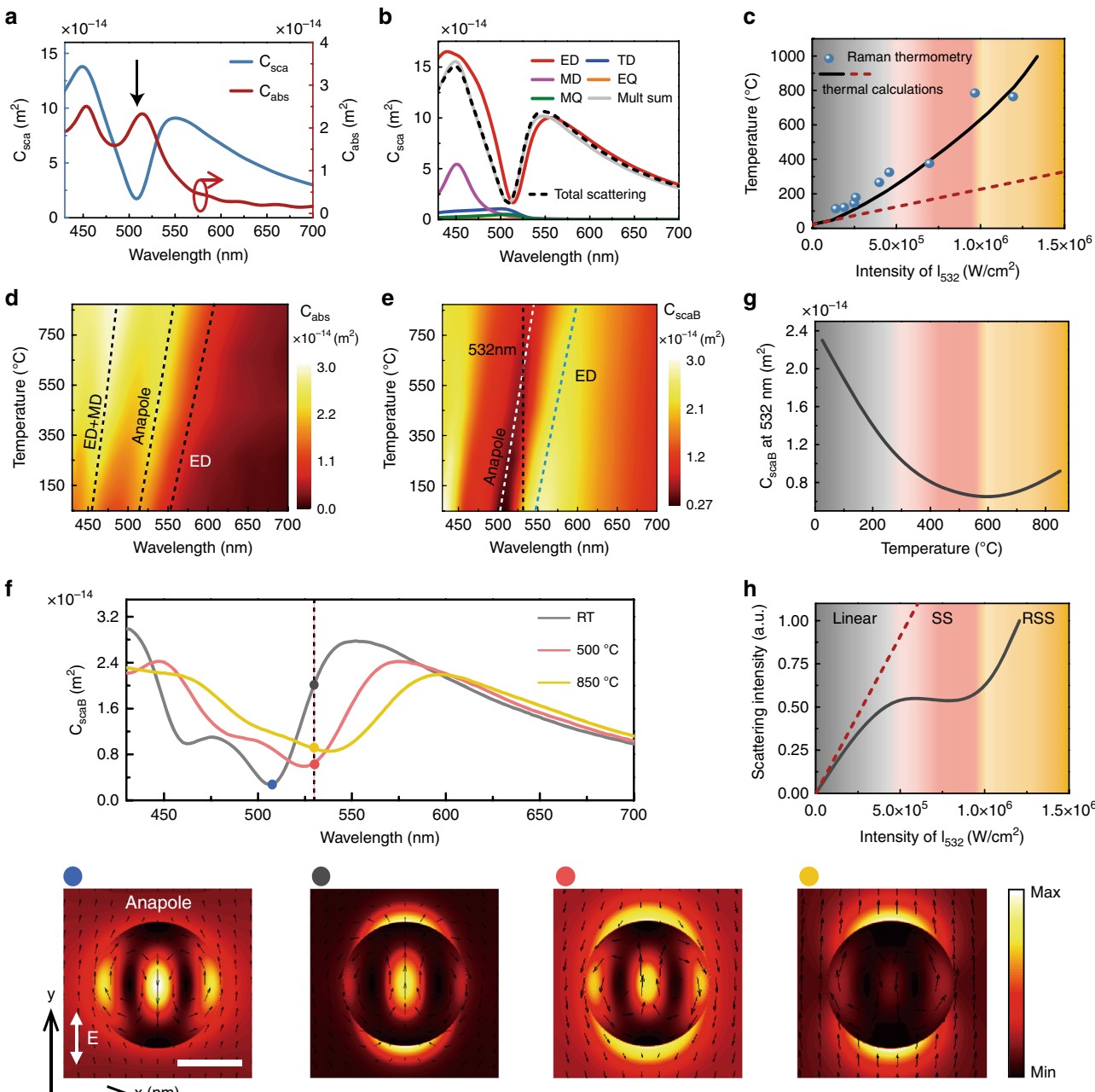

**Fig. 3 Anapole-driven photothermal nonlinearity. a** Simulations of optical scattering and absorption spectra of Si nanodisks. Gray arrow indicates the wavelength of the anapole mode featured with the scattering dip in the far-field and boosted absorptions. **b** Multipolar decomposition of induced currents in Cartesian coordinates. Mult sum is the sum of the scattering contributions of considered multipoles. **c** Temperature rises in Si nanodisks as a function of the intensity of 532 nm excitation beam. Dots represent extracted temperatures through Raman nanothermometry based on the intensity ratio of anti-Stokes and Stokes signals. Solid lines denote thermal calculations based on iterative algorithms (see Method). The linear trend shown in red dashed lines represents calculated temperature without taking the change of complex indices into account. Simulation maps of the absorption cross section (**d**) and backward scattering cross section (**e**) of Si nanodisks as a result of temperature rises. The dashed lines in **d** indicate a series of absorption maxima associated with corresponding dominant modes. The white, blue, and black dashed lines in **e** denote the anapole state, electric dipole state, and excitation wavelength, respectively. **f** Backward scattering cross section at three representative temperatures (top panel) and corresponding near-field distributions at the excitation wavelength of 532 nm (bottom panel). The white arrow in the field distribution indicates the incident light linearly polarized in the y-direction. Scale bar, 100 nm. **g** Backward scattering cross section at the wavelength of 532 nm at elevated temperatures, showing that it undergoes firstly suppression and then recovery during the photothermal tuning. **h** Calculated photothermal nonlinear scattering as a function of irradiance intensities.

shown in Fig. 4a. The outer contour of the RSS image can be subtracted over the SS image with *r* to be the subtractive factor, leaving a clear and sub-diffraction spot in the image. The 41-nm FWHM represents a localization capability that is far smaller than

the size of the nanodisk itself. To demonstrate that the localization accuracy works for not only isolated nanostructures, but also densely packed ones, we fabricated periodic Si nanodisk arrays with diameter of 200 nm, height of 50 nm, and pitch size of 300

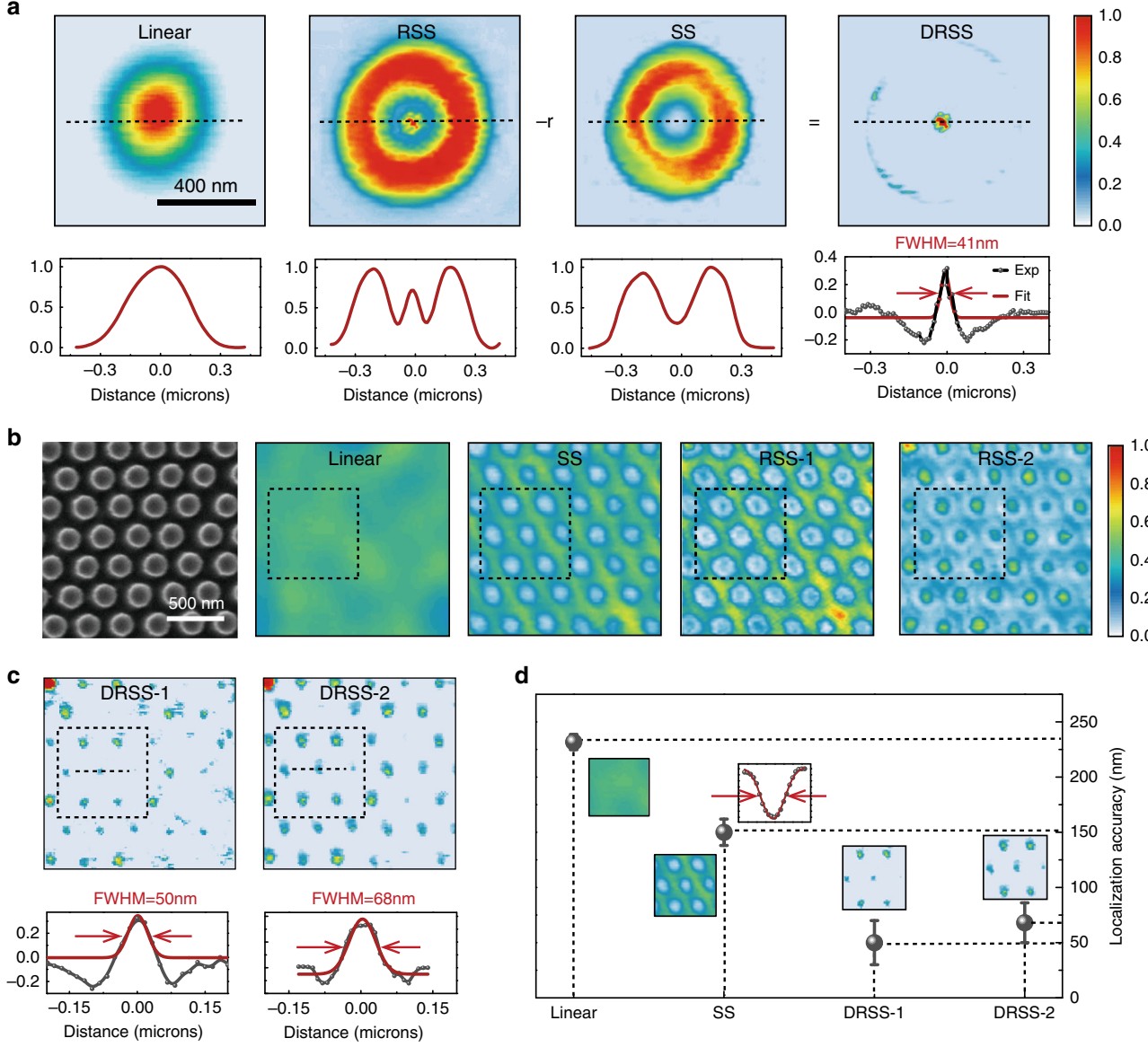

**Fig. 4 Optically localizing Si nanodisks packed in dense arrays based on the photothermal nonlinear scattering. a** PSF of a single isolated Si nanodisk at different irradiance intensities. Differential image (denoted as DRSS) between RSS and SS yields a localization precision with fitted FWHM of 41 nm with $r$ to be a subtractive factor. **b** PSFs of nonlinear scattering from periodic Si nanodisk arrays evolve with increasing excitation intensities. Correlated SEM image is also presented. The dense array of Si nanodisks in conventional confocal image is undistinguishable at the low excitation intensity. SS image generates a negative contrast whilst in RSS a central spike emerges. The onset of RSS process is presented in image RSS-1. When the degree of RSS increases, the central spike becomes more obvious (RSS-2). **c** Far-field optical localization of Si nanodisk arrays by means of differential image between RSS-1/RSS-2 and SS. **d** Localization accuracy scaling as PSFs obtained at different stages of nonlinear scattering. The error bars represent the deviations of FWHM values from 39 nanodisks in the scanning frame.

nm, as shown in Fig. 4b (Supplementary Fig. 10 for AFM characterization). The confocal image at low excitation intensities for such nanodisk array is blurry without any surprise. When gradually increasing the laser power, the evolution of PSFs unveils characteristic nonlinear scattering behaviors from SS to RSS (see Supplementary Note 11, and video for more results). By applying the differential technique, densely packed Si nanodisk arrays, whose density is beyond the diffraction limit can be distinctly localized (Fig. 4b). Combination of SEM and optical images unambiguously correlates the nonlinear scattering images with individual nanodisk morphology. Figure 4c quantifies the localization accuracy of, respectively, subtracting RSS-1 and RSS-2 to

SS, showing FWHMs below the diffraction limit in our experimental setup.

The localization accuracy of Si nanodisks acquired at different stages is plotted in Fig. 4d. The best localization accuracy corresponds to the very beginning of RSS while the signal to noise ratio is barely satisfactory. Strong RSS can induce better contrast while sacrificing a bit of precision. Noteworthily, during the whole photothermal nonlinearity modulation processes, the irradiance range is far from thermal deformation of Si nanodisks and the localization imaging is highly reproducible (Supplementary Fig. 12). We envision that the demonstrated photothermal nonlinearity assisted label-free

imaging modality could be potentially useful for contactless inspection and metrology of silicon ICs or failure analysis of microelectronic circuitry[52]. It is noted that the proposed technique is also applicable for other shapes and sizes of silicon nanostructures supporting anapole modes, not limited to nanodisks (Supplementary Fig. 13). In analogy to gold nanoparticles[35,37], the applicability of such Si nanostructures as fluorescence-free labeling contrasts in photothermal nonlinearity assisted cellular imaging might be feasible combining their biodegradable features.

## Discussion

In summary, we have demonstrated giant photothermal nonlinearity and active scattering modulation by fully exploiting the near- and far-field properties of anapole states in a single Si nanodisk. Taking advantages of the resonantly enhanced near-field absorption at the anapole's excitation, we observed pronounced temperature rises along with record-high refractive index changes under mild laser irradiances. Utilizing low-radiating feature of anapole modes, far-field scattering was dynamically controlled by photothermally tuning anapole spectral positions, allowing for active scattering engineering with all-optical stimuli. The anapole-driven photothermal nonlinear scattering results in dramatically changed PSFs in confocal reflectance images, offering the potential for localization of Si nanostructures with accuracy below the diffraction limit. As a proof-of-principle demonstration, densely packed Si nanodisk arrays are resolved with 40–60 nm FWHM, corresponding to $\lambda/10$ precision. Considering the compatibility with the existing semiconductor fabrication infrastructure, our work provides new perspectives for Si photonics with giant optical nonlinearity and the long-sought active control capability.

## Methods

**Preparation of silicon nanodisks.** Amorphous Si was deposited onto the glass substrate by magnetron sputtering. Then, polystyrene (PS) spheres were firstly spin-coated on to the layer film consisting of sputtered silicon onto the glass. The size of the PS mask was reduced by the RIE process using oxygen gas. And then such PS spheres serve as the mask for the subsequent fluorine-based inductively coupled plasma reactive ion etching (ICP-RIE) using CHF3 gas. Finally, the PS mask was removed with sonication in acetone. The sizes of the resulting silicon disks can be precisely tuned by changing the size of the PS mask with accurate control of the etching time. When fabricating large array of periodic Si nanodisks, self-organized PS spheres assembling in a hexagonally close-packing manner were prepared as the monolayer mask. The as-prepared Si samples are pre-annealed to switch into crystalline phase, before performing all the nonlinear scattering measurements (Supplementary Note 1).

**Thermal calculations.** The temperature growth inside the Si nanodisk is related to the absorbed power $Q = \sigma_{abs}I$ according to[53,54]

$$\Delta T = \frac{\sigma_{abs}I}{4\pi R_{eq}\kappa\beta} \tag{1}$$

where $\kappa$ is the thermal conductivity of the surrounding medium. In the present case for Si nanodisks on the glass substrate and immersed in the oil environment, $\kappa$ was taken to be 0.38. $\beta$ is a dimensionless geometrical correction factor for a geometry with axial symmetry. For Si nanodisks with D/h = 4, it is expressed as $\beta = \exp\{0.04 - 0.0124\ln 4 + 0.0677\ln^2 4 - 0.00457\ln^3 4\} = 1.15$. $R_{eq}$ is the corresponding equivalent radius, calculated as the radius of a sphere with the same volume as the nanodisk. The temperature rising from initial RT (25 °C) to the final temperature was divided into several intermediate steps and for each iteration, temperature-dependent optical absorption was firstly determined (Fig. 3d) and substituted into the formula Eq. (1). The derived temperature rise by photothermal effects shows a nice agreement with the results from Raman measurement. The linear trend shown in the red dashed line in Fig. 3c represents temperature rises linearly with irradiance intensities, providing an underestimation of the actual temperature without taking into account photothermal refractive index change.

**Complex refractive index of Si at elevated temperatures.** The present work focuses on the visible region, at which Si is generally more absorbing at photon energies close to the band gap. One has to consider both real and imaginary

parts of the refractive index. Values of $n$ and $k$ used for simulations were taken from the model given by Jellison[39] (see Supplementary Note 6). The real part $n$ was reported to vary linearly with temperature rise while the imaginary part $k$ varied exponentially with temperature[39,55]. Measurements of temperature dependence of $n$ and $k$ were performed up to 400 °C by ellipsometric techniques, which show good congruence with the model adopted from literature[39,40]. And then, an extrapolation was made to determine the complex refractive index at high temperatures.

**Microscope system.** The nonlinear scattering measurements were performed based on Abberior 775 STED confocal microscope (Abberior Instruments GmbH, Göttingen). We coupled continuous-wave laser line (532 nm) into the system for CW illumination[35,37]. The excitation beam was first spatially filtered and then focused onto the sample. Linear polarization excitation was controlled by imposing a half-wave plate on the laser beam. The backward scattering signal was collected using the same objective lens (×100, NA = 1.4, Olympus), reflected by a beam splitter and detected by a photomultiplier tube (PMT) after a confocal pinhole. The laser beam at the wavelength of 532 nm is focused by an objective (NA = 1.4) down to a diffraction-limited spot (the full width at half maximum ∼230 nm). Given a power of 2.11 mW reaching the sample, it yields an average intensity of 1.25 MW/cm². Under such circumstance, the disk raises its temperature to cause the refractive index change of 0.5. The corresponding absorbed power per disk is estimated to be 0.2–0.4 mW, and the estimated absorption efficiency is 9.5–19%. The microscope images were obtained by synchronizing the PMT and the galvo mirror scanner and were recorded by beam scanning through the sample with a step size of 7 nm and a dwell time of 10 μs.

**Multipole decomposition.** The Cartesian electric and magnetic dipole, quadrupole moments and the toroidal dipole moments of a nanodisk were calculated using the standard expansion formulas[10]:

*Electric dipole moment*:

$$\mathbf{P}_{car} = \frac{1}{i\omega}\int \mathbf{J}d^3\mathbf{r} \tag{2}$$

*Magnetic dipole moment*:

$$\mathbf{M}_{car} = \frac{1}{2c}\int (\mathbf{r}\times\mathbf{J})d^3\mathbf{r} \tag{3}$$

*Electric quadrupole moment*:

$$\mathbf{Q}^{\mathbf{E}}_{\alpha,\beta} = \frac{1}{i2\omega}\int [r_\alpha J_\beta + r_\beta J_\alpha - \frac{2}{3}\delta_{\alpha,\beta}(\mathbf{r}\cdot\mathbf{J})]d^3\mathbf{r} \tag{4}$$

*Magnetic quadrupole moment*:

$$\mathbf{Q}^{\mathbf{M}}_{\alpha,\beta} = \frac{1}{3c}\int [(\mathbf{r}\times\mathbf{J})_\alpha r_\beta + (\mathbf{r}\times\mathbf{J})_\beta r_\alpha]d^3\mathbf{r} \tag{5}$$

*Toroidal dipole moment*:

$$\mathbf{T}_{car} = \frac{1}{10c}\int \left[(\mathbf{r}\cdot\mathbf{J})\mathbf{r} - 2r^2\mathbf{J}\right]d^3\mathbf{r} \tag{6}$$

where $\mathbf{J} = i\omega\varepsilon_0(\varepsilon_r - 1)\mathbf{E}$ is the induced current in the structure, $\mathbf{r}$ is the position vector with the origin at the center of the nanodisk, and $\alpha, \beta = x, y, z$.

**Raman spectroscopy.** Raman spectra were taken under a microspectroscopic system based on an inverted optical microscope (NTEGRA Spectra, NT-MDT)[56]. Briefly, Si nanodisks were excited using linearly polarized 532-nm laser beams using an oil immersion objective (1.4 NA, ×60, Olympus). The resulting Raman signal with both Stokes and anti-Stokes lines was collected using the same objective, passed through a notch filter, and focused into the spectrometer with a cooled CCD (iDdus, Andor). Raman spectra were recorded with an acquisition time of 1 s.

## Data availability

The data that support the plots within this paper and other findings of this study are available from the corresponding author upon reasonable request.

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

## Acknowledgements

This research was supported by National Key R&D Program of China (2018YFB1107200), National Natural Science Foundation of China (NSFC) (61805107, 61905097, 61975067), Guangdong Provincial Innovation and Entrepreneurship Project (Grant 2016ZT06D081), Natural Science Foundation of Guangdong Province (2017A030313006). S-W.C is funded by Ministry of Science and Technology, Taiwan (105-2628-M-002 -010 -MY4 and 108-2321-B-002 -058 -MY2). S-W.C acknowledges the support by MOST and Ministry of Education, Taiwan (MOE) under The Featured Areas Research Center Program within the framework of the Higher Education Sprout Project.

## Author contributions

X.L. and S.-W.C. conceived the idea and supervised the project. T.Z. and Y.C. performed the experiments. K.C. prepared the sample. T.Z. and Y.X. performed the electromagnetic multipolar expansion. J.X and J.H assisted dark-field experiments. T.W., G.L., B.W., and X.X contributed to Raman measurements. Y.S.D. and Y.L.T. assisted ellipsometric spectroscopy measurements. X.L. and Y.Y.C. contributed to analysis of point spread functions of scattering images. T.Z., Y.C., S.-W.C., and X.L. analyzed data and prepared the manuscript. T.Z., Y.C., B.G., S.-W.C., and X.L. participated in the discussion and manuscript writing.

## Competing interests

The authors declare no competing interests.
