## [Peer Review File · Nature Communications]

Reviewers' comments:

Reviewer #1 (Remarks to the Author):

In their paper entitled "Anapole Mediated Giant Photothermal Nonlinearity for Super-Localization Nanoscopy," Zhang et al. report on giant modifications of the optical response (scattering) of silicon nanoparticles that support anapole-type resonances. They make use of the laser field localization with a nanoparticle resonance to promote optical absorption that translates into the thermo-optic effect that causes the spectral shift of the resonance and scattering cross section modification. I think the results the authors present are interesting and cutting-edge, and the approach to induce high modulation depths in the optical response that employs the anapole (=low-scattering) mode is very promising. However, I think that, in its present form, the manuscript is rough and requires substantial work before it can be accepted for publication in Nature Communications. Here are some of the issues I found after carefully reading the work of Zhang et al.:

1. The mentions of superresolution microscopy in this paper are misleading. The authors indeed observe subwavelength features in the (scattering) images of the sample through photothermal modifications of the near-field response. However, conventionally, in superresolution microscopy, the main objective is to image an unknown object of study, such as cell organelles, and increase the resolution of such imaging to beat the diffraction limit. Unless the authors can clearly demonstrate their approach to obtain superresolution images from other objects, not the silicon disks themselves, the mentions of superresolution should be removed from the manuscript.
2. Stylistically, I would like to strongly encourage the authors to reduce the degree of emotion when describing their results. As an example, in the abstract, the words "unparalleled," "exotic," "extraordinary," "tremendous," and "unprecedented" can be removed without modifying the message. I urge the authors to clear the manuscript from the unnecessary modifiers; they make reading the paper very hard. Additionally, there are multiple cases of article misuse and single-plural verb forms inconsistencies. I strongly recommend reaching out to an English-editing service to proof-edit the manuscript before resubmission.
3. Line 86: what is a negative scattering cross section? What are the units here? Also, it is not correct to use the figure of 200% to characterize a relative value that passes through zero.
4. Line 83: the intensity value is missing.
5. Lines 139-140: It is unclear how 200% was obtained from the values of 1, 0.1 and 1.1.
6. Line 160: it is unclear, how the Δn of 0.5 was obtained. In Ref. 40, chapter III is devoted to photothermal measurements of optical constants (erroneously called "Thermo-optic effects" in the reference list) and does not clarify the method of determining the thermo-optic tuning measurement.

Reviewer #2 (Remarks to the Author):

Manuscript 'Anapole Mediated Giant Photothermal Nonlinearity for Super-Localization Nanoscopy' reports about demonstration of large thermally induced change of the Si refractive index (~ 0.5) by utilizing high absorption at anapole state in Si discs. It also proposes to use this effect for high-resolution microscopy, which is quite arguable (please see my comment below). The paper is well written, with satisfactory level of English (this level is excellent in the abstract and introduction, but in the rest of the text there are several mistakes and misspellings). The results are clearly presented and well-analyzed from different perspectives. I believe, it is suitable to be published in Nature Communications, after the following major and minor comments are considered.

Major comment: authors claim to use a changeable PSF for super-localization nanoscopy (Figure 4 and corresponding text), with which I cannot agree. The demonstrated technique allows super-resolution determination of the position of the Si disc, and it has no use for any other imaging (for example, of organic cells, etc.) And even for the first purpose, a simple method exists: fit

Gaussian-like PSF with a Gaussian function and extract its center as fitting parameters. Of course, this works only for isolated discs like in Figure 4a, and it won't work for a dense array as in Figure 4b,c (which is just slightly subwavelength, with a periodicity of 300 nm at the wavelength of 532 nm). However, this is still just imaging of Si discs, and I don't think this is a useful application. Maybe I don't understand properly how the proposed technic can be useful; otherwise authors can just drop this point – I do believe that the demonstration of huge thermal rise in Si discs and its investigation from different sides (including temperature measurements by Raman spectroscopy) is a large enough reason for this work to be published in Nature Communications.

Minor comments:

1) In the text authors mention intensity, but how exactly was it defined? I mean, the illumination spot has Gaussian distribution as shown in figure S6b. Therefore, the mentioned intensity – is it the peak intensity in the center of the spot, or is it kind of average intensity? Additionally, it would be nice to see the total laser power used, for example, to create the intensity of 1.3 MW/cm^2 (i.e., the one to reach the anapole state) – this might be useful for others to estimate, whether they can reproduce the effective with their setup.

2) I miss the description of used Si – was it crystalline (e.g., from SOI substrate) or amorphous (for example, made by physical vapor deposition, PVD)?

3) If amorphous Si was used, then it is known that high temperature can lead to crystallization. It would be nice to see some discussion about it.

4) What material properties were used for simulations of Si? I mean the refractive index itself (for 532 nm and for the whole spectrum) and its dependence on the temperature.

5) Regarding the dependence of n_{Si} on the temperature: authors give two references, one for Palik (line 159 in the main text), and another to [G. E. Jellison Jr. and F. A. Modine] (Supplementary Note 2, and reference 2 in the supplementary). However, these two references don't match: Palik gives non-linear temperature dependence for crystalline Si as $dn/dT = B_1 + B_2T + B_3T^2$ (though, values for B coefficients are given for IR wavelengths), while the second reference assumes linear dependence. Therefore, could authors clearly state which model they have used? By the way, the second reference in the supplementary has a mistake: the last name of the first author (Jellison) for some reason was shortened to one letter. Finally, there is also a reference 31 in the main text (mentioned in line 99), which also reports about non-linear thermo-optic effect.

6) I think, both references 1 and 2 from Supplementary are quite important for the work (the first describes how to measure temperature by Raman spectroscopy, and the second is about the thermo-optic coefficient of Si). Therefore, I believe, they should be mentioned in the text, and maybe even the used technics can be mentioned, for example, in Methods.

7) I also miss a description of objectives used for laser scanning setup and dark-field microscopy (Figure S1b).

8) Line 86: 'The normalized scattering cross-sections can be reversibly regulated from -0.9 to 1.1...' – the negative number for the scattering cross-section sounds weird; therefore, it is hard to catch, what authors meant here (only after reading the whole manuscript it becomes clear). I would recommend to re-phrase this sentence.

9) Line 114: 'The anapole state is an engineered superposition of toroidal and electric multipoles...' – well, this is not quite exact. Both toroidal and electric dipoles can be non-zero, but anapole state appears when they interfere destructively (also mentioned in line 62). I would rather rewrite this sentence as 'The anapole state is an engineered destructive interference between toroidal and electric dipoles...'

10) Line 163: 'Compared with the photothermal induced index change $\Delta n \sim 3 \times 10^{-5}$ in bulk Si under the same laser excitation intensity⁴¹,...' This is not a fair comparison, because in reference 41 the used wavelength was telecom, at which there is no absorption in Si. What would be nice to compare with is the photothermally induced index change for bulk Si (or for 50-nm-thin Si film used by authors) using the same setup and wavelength.

11) Figure 1b: To support the idea of linear scattering at small intensities, authors could add a fitting straight line (similarly to red dashed line in figures S5 a, d, e, h).

12) Figure 2e: I am wondering, why for small illumination intensities the normalized cross-section is not 1 exactly? Is it because of large errors for small intensities?

- 13) Figures 2-4: I am wondering, why blue-white-red colorbar was used in so many images? I mean, usually this colorbar is used for plotting signed values (with zero being white color, positive values as red, and negative as blue). For unsigned maps like Figure 2d a more suitable is a rainbow colormap (like the one in Figure S6), with zero (or min) being darkest and max being brightest colors. It is a matter of taste, so I leave the final choice to authors.
- 14) Figure 3 and, maybe, other figures: the text is too small to read (especially in panel a).
- 15) Figure 3b: The multipole decomposition results look strange. Was it done for Si disk on a glass substrate or in vacuum? It is because formulas mentioned in corresponding Methods section are valid for even dielectric environment (and it is hard to do it in uneven surrounding). Therefore, other scientists usually do additional simulations for nanoparticles in even surrounding (for example, air), which gives qualitative image of multipoles. Then, when substrate is added, it can introduce small shifts of the resonances, but it won't drastically change the multipole combination. Secondly, scientists usually calculate contributions of multipoles to the total scattering cross-section (the formula can be taken, for example, from <https://doi.org/10.1021/acs.nanolett.7b04200>, see Methods section there). Then multipole contributions and their sum are plotted together with the total scattering cross-section (as example, see Figure S1 in mentioned reference). By comparing the sum of multipole contributions with the total scattering cross-section, one can judge whether enough multipole orders are considered.
- 16) Authors mention their modification as reversible and repeatable, with Figure 2f as one of the proofs. Another such proof could be dark-field measurements of the spectra before and after the heating (I mean, adding the last to the Figure S1b).
- 17) Supplementary Note 1: it would be great to see a final equation for the temperature as a function of anti-Stokes-to-Stokes ratio. Also, the phonon energy $h\Omega$ – was it measured or used as a fitting parameter?
- 18) Supplementary line 107: n_0 should not depend on T (i.e., ' $n(E,T) = n_0(E) + \dots$ ')
- 19) Supplementary Figure S4: why sum of C_{scaF} and C_{scaB} does not equal to the total C_{sca} ?
- 20) Supplementary Figure S5: first, the panels are small and a hard to read. Secondly, to simplify the comparison between three disks, it is better to combine plots: S5a, e, and Figure 3c into one plot; S5 c, g, and Figure 3g into another plot, and the same for S5 d, h, and Figure 3h. Finally, why the line and the linear fit in Figure S5d coincides? According to S5c, the scattering cross-section decreases with temperature, therefore it should not be the same straight line as for small intensities.
- 21) Supplementary Figure S6a: I am wondering, why authors decided to fit the dependence with five order polynom? I don't see it being used anywhere else.
- 22) Supplementary Figure S6c-f: why all panels in the top show diagonal symmetry? And what was the polarization? Finally, what does these images show?
- 23) Supplementary Figure S7a, right panel: why some holes, according to the colormap, are below 10 nm, when the max is 60 nm (top of disks), and disk height is 50 nm? Is it because glass substrate was also etched between disks?
- 24) As I mentioned in the beginning, there are several mistakes and misspellings in the main text. Here are some examples:
- Line 132: 'Further increases of excitation intensities, a sharp peak emerges...' – I believe, it should be something like 'By further increasing the excitation intensity, a sharp peak emerges...'
 - Line 305: 'Linear polarization excitation was obtained by imposing a half-wave plate on the laser beam.' – Half-wave plate does not give linear polarization; it can only rotate it. I think, authors meant here 'Linear polarization excitation was controlled by imposing a half-wave plate on the laser beam.'
 - Line 330: 'Raman spectra were recorded over an acquirement time of 1 s.' I think, usually this is called acquisition time.
 - Line 471: 'Illustration of strong optical heating that efficientLY converts...'

I hope I expressed myself clearly, but if there is any question regarding my comments, authors are welcome to contact me for clarification.

Sincerely,
Vladimir Zenin

Manuscript ID: NCOMMS-20-05933

Manuscript title: “Anapole Mediated Giant Photothermal Nonlinearity in Nanostructured Silicon”

Point-by-point responses to Reviewers' Comments

We are very grateful for all the comments from the editor and all the reviewers. These comments are very important and valuable to improve the quality and readability of this paper. Revisions and responses to address your comments are presented as below.

#####

Reviewer #1 (Remarks to the Author):

Comment 1) In their paper entitled “Anapole Mediated Giant Photothermal Nonlinearity for Super-Localization Nanoscopy,” Zhang et al. report on giant modifications of the optical response (scattering) of silicon nanoparticles that support anapole-type resonances. They make use of the laser field localization with a nanoparticle resonance to promote optical absorption that translates into the thermo-optic effect that causes the spectral shift of the resonance and scattering cross section modification. I think the results the authors present are interesting and cutting-edge, and the approach to induce high modulation depths in the optical response that employs the anapole (=low-scattering) mode is very promising.

Reply: We highly appreciate the reviewer’s positive comment and acknowledging the novelty of our work as interesting and cutting-edge results.

Comment 2) However, I think that, in its present form, the manuscript is rough and requires substantial work before it can be accepted for publication in Nature Communications. Here are some of the issues I found after carefully reading the work of Zhang et al.:

Reply: As shown in the following responses, we carefully re-examined and re-analyzed the data that would induce possible ambiguity previously. We hope that the English editing, revised figure presentations, additional experiments and revised

data interpretation have addressed the reviewer's concerns, making this work stronger and fitting the high standards of Nature Communications.

Comment 3) The mentions of superresolution microscopy in this paper are misleading. The authors indeed observe subwavelength features in the (scattering) images of the sample through photothermal modifications of the near-field response. However, conventionally, in superresolution microscopy, the main objective is to image an unknown object of study, such as cell organelles, and increase the resolution of such imaging to beat the diffraction limit. Unless the authors can clearly demonstrate their approach to obtain superresolution images from other objects, not the silicon disks themselves, the mentions of superresolution should be removed from the manuscript.

Reply: We thank the reviewer for the constructive comments. We agree that currently the super-localization is only on the Si nanodisks and nowadays super-resolution microscopy aims to observe an unknown object beyond the diffraction limit. To avoid misleading, we have taken the reviewer's suggestion and removed superresolution microscopy throughout the manuscript.

Nevertheless, we would like to emphasize that the demonstrated technique has huge potentials in relevant fields. *Firstly*, this is a label-free modality of silicon nanostructures. Conventional fluorescence-based super-resolution techniques (such as STORM, PALM, STED, etc.) are all contrast agents dependent. The demonstrated photothermal nonlinearity assisted non-invasive imaging could be potentially useful for contactless inspection and metrology of silicon ICs or failure analysis of micro-electronic circuitry. *Secondly*, our proposed technique is also applicable for other shapes and sizes of silicon nanostructures, not limited to nanodisks. The simulations in Figure R1 showcase that the anapole state can be supported in Si nanospheres of different sizes at different wavelengths. In analogy to gold nanoparticles [1-3], the applicability of such Si nanospheres as fluorescence-free labeling contrasts in photothermal nonlinearity assisted super-resolution cellular imaging might be feasible combining their biodegradable features.

Please see changes and additional discussions in Pages 9 and 10 to envision the potential applications.

Fig.R1 Simulation map of total scattering spectra of Si nanospheres with various sizes. Dash lines indicate the scattering valleys (anapole state) varying with diameters of nanospheres.

References

- [1]. Xu, J. et al. Plasmonic Nanoprobes for Multiplexed Fluorescence-Free Super-Resolution Imaging. *Adv Opt Mater* **6**, 1800432 (2018).
- [2]. Ouyang, X. et al. Invited Article: Saturation scattering competition for non-fluorescence single-wavelength super-resolution imaging. *APL Photonics* **3**, 110801 (2018).
- [3]. Wu, H. Y. et al. Ultrasmall all-optical plasmonic switch and its application to superresolution imaging. *Sci Rep* **6**, 24293 (2016).

Comment 4) Stylistically, I would like to strongly encourage the authors to reduce the degree of emotion when describing their results. As an example, in the abstract, the words “unparalleled,” “exotic,” “extraordinary,” “tremendous,” and “unprecedented” can be removed without modifying the message. I urge the authors to clear the manuscript from the unnecessary modifiers; they make reading the paper very hard. Additionally, there are multiple cases of article misuse and single-plural verb forms inconsistencies. I strongly recommend reaching out to an English-editing service to proof-edit the manuscript before resubmission.

Reply: We have revised the manuscript thoroughly following the reviewer’s suggestions. We have sought English editing services to improve the readability. The

manuscript has been extensively polished to reduce the degree of emotion and to avoid the grammar mistakes and misspellings. References have been re-examined.

Comment 5) Line 86: what is a negative scattering cross section? What are the units here? Also, it is not correct to use the figure of 200% to characterize a relative value that passes through zero.

Reply: We thank the reviewer for professional comments. These numbers mentioned here represent the modulation depth of normalized scattering cross sections. To avoid any confusions, the corresponding sentence has been re-phrased as: “The scattering cross-section that is normalized to 1 for linear scattering can be reversibly suppressed down to 0.1 and then rapidly enhanced up to 1.1, demonstrating a large modulation depth and a broad dynamic range, due to the progressive transition of dominant modes from the bright state to the low-radiating dark state and further moving towards the bright state again.” (1st paragraph, page 4 in the revised text)

The normalized scattering cross-section mentioned here (also in Fig.2e) has arbitrary unit. It is taken from the ratio of scattering over excitation intensity, and then normalized to unity for linear scattering region. Please see changes in Page 4 and 6.

Comment 6) Line 83: the intensity value is missing.

Reply: In general cases of optical nonlinearity, the operating laser intensity is on the level of \sim GW/cm² (ref. 4 and 5). The specific value was omitted here to illustrate an alleviated intensity level required to achieve similar nonlinearity. Following the reviewer’s suggestion, the sentence has been re-phrased as “A record-high photothermal refractive index change Δn up to 0.5 can be achieved upon a mild laser radiance intensity of 1.25 MW/cm² through optically pumping a Si nanodisk at the wavelength close to the anapole mode”. (1st paragraph, page 4 in the revised text)

[4]. Shcherbakov, M. R. et al. Ultrafast All-Optical Switching with Magnetic Resonances in Nonlinear Dielectric Nanostructures. *Nano Lett* **15**, 6985-6990 (2015).

[5]. Alam, M. Z., Schulz, S. A., Upham, J., De Leon, I. & Boyd, R. W. Large optical nonlinearity of nanoantennas coupled to an epsilon-near-zero material. *Nat Photonics*

12, 79-83 (2018).

Comment 7) Lines 139-140: It is unclear how 200% was obtained from the values of 1, 0.1 and 1.1.

Reply: The modulation depth of normalized scattering cross sections is from suppressed value of 0.1 to enhanced value of 1.1. To remove the confusion, we have clarified and re-phrased the sentence: “As shown in Fig.2e, the normalized scattering cross-section stays constant (normalized to 1) at the initial linear region, then decreases to 0.1 for the largest SS, and again drastically rises to 1.1 for RSS, demonstrating a large dynamic range spanning from scattering suppression to enhancement.” (2nd paragraph, page 6 in the revised text)

Comment 8) Line 160: it is unclear, how the delta-n of 0.5 was obtained. In Ref. 40, chapter III is devoted to photothermal measurements of optical constants (erroneously called “Thermo-optic effects” in the reference list) and does not clarify the method of determining the thermo-optic tuning measurement.

Reply: We thank the reviewer for this very important comment. The delta-n was obtained through the following strategy:

First, we took Raman scattering spectra to extract the temperature rise of Si nanodisks under different illumination intensities. A pronounced temperature rise from room temperature to 950°C was found among the illumination intensities used in the experiments. *And then*, refractive index of Si was extrapolated based on its temperature dependence following the empirical expressions given by the article from Jellison and Modine [6], in which n and k are both parameterized as functions of temperature T and photon energy E for photon energies below the direct band edge of silicon. Although Jellison’s work only gave measurements up to 490 °C, the extrapolation was further corroborated with experimental data at 500°C, 700°C and 850°C measured by Šik et al [7] (Fig.R2), validating its soundness.

In addition, we supplemented both temperature-dependent index literatures and refractive index measurements of a Si thin film of 170 nm thickness by ellipsometric technique up to 400°C which is limited by the instrument. The measured data agree

well with the reported ones and an extrapolated refractive index change reaching about 0.5 is confirmed. Such revealed substantial refractive index change is responsible for the photothermal induced Mie resonance shifts, which was corroborated by the giant modification of the scattering and the PSFs.

After close scrutiny of the formula in reference 40 in which the proposed refractive index nonlinear dependence on temperature is only applicable to transparent sample without absorption, not corresponding to the case in the present work, therefore we have substituted by refs. 39, 40 in the main text accordingly (1st paragraph, page 7). We have also added a short description of the model about the complex refractive index of Si at elevated temperatures in the Methods sections.

Fig.R2 Temperature-dependent complex refractive index of Si at the wavelength of 532 nm from room temperature to 1000°C. Squares, stars and circles denote data given by Jellison et al [ref. 6], Šik et al [ref. 7] and our own measurements, respectively. The dash lines are the extrapolation based on the existing data.

[6]. Jellison, G. E. & Modine, F. A. Optical functions of silicon at elevated temperatures. *Journal of Applied Physics* **76**, 3758-3761 (1994).

[7]. Šik, J., Hora, J. & Humlíček, J. Optical functions of silicon at high temperatures. *Journal of Applied Physics* **84**, 6291-6298 (1998).

#####

Reviewer #2:

Manuscript ‘Anapole Mediated Giant Photothermal Nonlinearity for Super-Localization Nanoscopy’ reports about demonstration of large thermally induced change of the Si refractive index (~ 0.5) by utilizing high absorption at anapole state in Si discs. It also proposes to use this effect for high resolution microscopy, which is quite arguable (please see my comment below). The paper is well written, with satisfactory level of English (this level is excellent in the abstract and introduction, but in the rest of the text there are several mistakes and misspellings). The results are clearly presented and well-analyzed from different perspectives. I believe, it is suitable to be published in Nature Communications, after the following major and minor comments are considered.

Reply: We deeply thank the reviewer for acknowledging the novelty and quality of our work and professional technical comments.

Major comment:

Comment 1) authors claim to use a changeable PSF for super-localization nanoscopy (Figure 4 and corresponding text), with which I cannot agree. The demonstrated technic allows super-resolution determination of the position of the Si disc, and it has no use for any other imaging (for example, of organic cells, etc.) And even for the first purpose, a simple method exists: fit Gaussian-like PSF with a Gaussian function and extract its center as fitting parameters. Off course, this works only for isolated discs like in Figure 4a, and it won’t work for a dense array as in Figure 4b,c (which is just slightly subwavelength, with a periodicity of 300 nm at the wavelength of 532 nm). However, this is still just imaging of Si discs, and I don’t think this is a useful application. Maybe I don’t understand properly how the proposed technic can be useful; otherwise authors can just drop this point – I do believe that the demonstration of huge thermal rise in Si discs and its investigation from different sides (including temperature measurements by Raman spectroscopy) is a large enough reason for this work to be published in Nature Communications.

Reply: We thank the reviewer for the constructive comments. We agree that currently the super-localization is only on the Si nanodisks and nowadays super-resolution microscopy aims to observe an unknown object beyond the diffraction limit. To avoid misleading, we have taken the reviewer's suggestion and removed superresolution microscopy throughout the manuscript.

Nevertheless, we would like to emphasize that the demonstrated technique has huge potentials in relevant fields. *Firstly*, this is a label-free modality of silicon nanostructures. Conventional fluorescence-based super-resolution techniques (such as STORM, PALM, STED, etc.) are all contrast agents dependent. The demonstrated photothermal nonlinearity assisted non-invasive imaging could be potentially useful for contactless inspection and metrology of silicon ICs or failure analysis of micro-electronic circuitry. *Secondly*, our proposed technique is also applicable for other shapes and sizes of silicon nanostructures, not limited to nanodisks. The simulations in Figure R1 showcase that the anapole state can be supported in Si nanospheres of different sizes at different wavelengths. In analogy to gold nanoparticles [refs. 1-3], the applicability of such Si nanospheres as fluorescence-free labeling contrasts in photothermal nonlinearity assisted super-resolution cellular imaging might be feasible combining their biodegradable features. *At last*, as already pointed out by the reviewer, fitting with a Gaussian function and extracting its center to super-localize the position only work on the isolated nanostructures. Our method demonstrates a high localization accuracy for imaging dense arrays of Si nanostructures.

Please see changes and additional discussions in Pages 9 and 10 to envision the potential applications.

Fig.R1 Simulation map of total scattering spectra of Si nanospheres with various sizes. Dash lines indicate the scattering valleys (anapole state) varying with diameters of nanospheres.

References

- [1]. Xu, J. et al. Plasmonic Nanoprobes for Multiplexed Fluorescence-Free Super-Resolution Imaging. *Adv Opt Mater* **6**, 1800432 (2018).
- [2]. Ouyang, X. et al. Invited Article: Saturation scattering competition for non-fluorescence single-wavelength super-resolution imaging. *APL Photonics* **3**, 110801 (2018).
- [3]. Wu, H. Y. et al. Ultrasmall all-optical plasmonic switch and its application to superresolution imaging. *Sci Rep* **6**, 24293 (2016).

Minor comments:

Comment 1) In the text authors mention intensity, but how exactly was it defined? I mean, the illumination spot has Gaussian distribution as shown in figure S6b. Therefore, the mentioned intensity – is it the peak intensity in the center of the spot, or is it kind of average intensity? Additionally, it would be nice to see the total laser power used, for example, to create the intensity of 1.3 MW/cm^2 (i.e., the one to reach the anapole state) – this might be useful for others to estimate, whether they can reproduce the effective with their setup.

Reply: The intensity mentioned in the text is the average intensity, unless otherwise

specified. The laser beam at the wavelength of 532 nm is focused by an objective (NA=1.4) down to a diffraction-limited spot (the full-width at half-maximum ~ 230 nm). Given a power of 2.11 mW reaching the sample, it yields an average intensity of 1.25 MW/cm^2 . A description about such experimental condition has been added in the revised manuscript in the Methods section.

Comment 2) I miss the description of used Si – was it crystalline (e.g., from SOI substrate) or amorphous (for example, made by physical vapor deposition, PVD)?

Reply: The as-prepared Si is amorphous and it was deposited by magnetron sputtering. We have added the description about the preparation of Si nanodisks in Methods.

Comment 3) If amorphous Si was used, then it is known that high temperature can lead to crystallization. It would be nice to see some discussion about it.

Reply: The reviewer is correct that the huge temperature rise in Si nanodisks can lead to crystallization. We also observed the laser crystallization when we increased the laser power and this was confirmed by ex-situ confocal Raman measurements. The amorphous silicon (a-Si for short) sample was found to undergo phase transformation when the laser power exceeded 0.5 mW (corresponding to 0.3 MW/cm^2). The appearance of a Raman peak at 520 cm^{-1} in the laser treated sample implies the onset of crystallization. On the other hand, for the a-Si nanodisks without laser annealing, the Raman spectrum contains only a broad band around 475 cm^{-1} . After laser annealing, the Si can be poly-crystalline (poly-Si) or single crystalline (c-Si). Considering the similar optical properties between poly-Si and c-Si [ref. 4], the refractive index of crystalline silicon has been used for all of our simulations.

Fig. R2 Raman scattering before and after the laser crystallization.

In our experiments, the as-prepared a-Si nanodisks were pre-treated with relative high laser power of 1.5 mW by fast scanning (dwell time $\sim 10\mu\text{s}$), leading to phase transformation to crystalline Si. After such laser treatment, the Si nanodisks became thermally stable, and then we started the nonlinear scattering measurements. The nonlinear scattering processes are fully repeatable and reversible which have been confirmed by scattering images and intensities (shown in Figs. 2d and f), dark field scattering spectra (Supplementary Figure S3), and SEM images (Supplementary Figure S11) before and after the measurements.

In response to the constructive comments 2 and 3, we have added a discussion in Supplementary Note 1 about the sample description to clarify the material properties.

[4]. Mirshafieyan, S.S. & Guo, J. Silicon colors: spectral selective perfect light absorption in single layer silicon films on aluminum surface and its thermal tunability. *Opt Express* **22**, 31545-31554 (2014).

Comment 4) What material properties were used for simulations of Si? I mean the refractive index itself (for 532 nm and for the whole spectrum) and its dependence on the temperature.

Reply: The refractive index of crystalline silicon was used in the simulation

throughout the manuscript. Its temperature dependence follows the empirical expressions given by the article from Jellison and Modine [ref. 5], in which n and k are both parameterized as functions of temperature T and photon energy E for photon energies below the direct band edge of silicon. Although Jellison's work only gave measurements up to 490 °C, the extrapolation was further corroborated with experimental data at 500°C, 700°C and 850°C measured by Šik et al [ref. 6] (Fig.R3), validating its soundness. In addition, we supplemented experimental measurements of temperature-dependent complex refractive index of single crystalline silicon with thickness of 170 nm. Restricted by the heater equipped in the ellipsometer, the measurement was up to 400°C. Our measurements show good congruence with these literatures.

To make it clear, we have clarified this part in Supplementary Note 5.

Fig.R3 Temperature-dependent complex refractive index of Si at the wavelength of 532 nm from room temperature to 1000°C. Squares, stars and circles denote data given by Jellison et al [ref. 5], Šik et al [ref. 6] and our own measurements, respectively. The dash lines are the extrapolation based on the existing data.

[5]. Jellison, G. E. & Modine, F. A. Optical functions of silicon at elevated temperatures. *Journal of Applied Physics* **76**, 3758-3761 (1994).

[6]. Šik, J., Hora, J. & Humlíček, J. Optical functions of silicon at high temperatures. *Journal of Applied Physics* **84**, 6291-6298 (1998).

Comment 5) Regarding the dependence of n_{Si} on the temperature: authors give two references, one for Palik (line 159 in the main text), and another to [G. E. Jellison Jr. and F. A. Modine] (Supplementary Note 2, and reference 2 in the supplementary). However, these two references don't match: Palik gives non-linear temperature dependence for crystalline Si as $dn/dt = B_1 + B_2T + B_3T^2$ (though, values for B coefficients are given for IR wavelengths), while the second reference assumes linear dependence. Therefore, could authors clearly state which model they have used? By the way, the second reference in the supplementary has a mistake: the last name of the first author (Jellison) for some reason was shortened to one letter. Finally, there is also a reference 31 in the main text (mentioned in line 99), which also reports about non-linear thermo-optic effect.

Reply: We thank the reviewer for the professional comments. Indeed, there are numerous references for examining the temperature dependence of refractive index of Si. Many of them concern the variation of the refractive index with the temperature for transparent optical materials, where the thermo-optic coefficient denoted as dn/dT has been investigated. In their studies, the optical communication wavelength of 1.5 μm or other IR region are considered since Si has almost no absorption. Taking into consideration that complex refractive index of Si is related to its energy gap which can be tuned with temperature, one can deduce a nonlinear temperature dependence of dn/dT , like mentioned in the book from Palik and ref. 31 in the previous main text. On the other hand, many studies focus on the visible region, at which Si is generally more absorbing at photon energies close to the band gap. One has to consider both real and imaginary part of the refractive index. Values of n and k are determined by fitting an optical model to the measured data. The real part n is found to vary linearly with temperature rise while the imaginary part k is exponentially increasing with temperature [ref. 7 & 8] (also ref. 5 and 6 in the response to comment 4).

For simulations in the present work, we used refractive index of silicon at elevated temperatures following the Jellison model (Supplementary Note 5 in the

revised manuscript). To make it clear and avoid misleading, the references in the main text have been replaced by those matching the model that we used in the revised manuscript (ref. 39 and 40 on page 7). The description in the supplementary information (see Supplementary Note 5) has also been revised. The mistake about the author name in the reference in supplementary information has been corrected.

[7]. Bergmann, J., Heusinger, M., Andrä, G. & Falk, F. Temperature dependent optical properties of amorphous silicon for diode laser crystallization. *Opt Express* **20**, A856-A863 (2012).

[8]. Hoyland, J. D. & Sands, D. Temperature dependent refractive index of amorphous silicon determined by time-resolved reflectivity during low fluence excimer laser heating. *Journal of Applied Physics* **99**, 063516 (2006).

Comment 6) I think, both references 1 and 2 from Supplementary are quite important for the work (the first describes how to measure temperature by Raman spectroscopy, and the second is about the thermo-optic coefficient of Si). Therefore, I believe, they should be mentioned in the text, and maybe even the used technics can be mentioned, for example, in Methods.

Reply: We have taken the reviewer's suggestions. The method we used to measure the temperature by Raman spectroscopy has already been mentioned both in the main text (last paragraph on page 6 and the ref. 38) and in Methods. In the revised manuscript, we have added a short description of the model about "complex refractive index of Si at elevated temperatures" in the Methods section.

Comment 7) I also miss a description of objectives used for laser scanning setup and dark-field microscopy (Figure S1b).

Reply: The laser scanning setup utilizes a high numerical aperture objective lens (Olympus, 100×, NA=1.4). The dark-field microscopy employs a dark-field objective lens (Olympus, MPlanFL, 50×, NA=0.8). We have added relevant information in the revised manuscript (see Methods-Microscope system and Supplementary Note 3).

Comment 8) Line 86: ‘The normalized scattering cross-sections can be reversibly regulated from -0.9 to 1.1...’ – the negative number for the scattering cross-section sounds weird; therefore, it is hard to catch, what authors meant here (only after reading the whole manuscript it becomes clear). I would recommend to re-phrase this sentence.

Reply: We have followed the reviewer’s suggestion and the sentence has been re-phrased as “The scattering cross-section that is normalized to 1 for linear scattering can be reversibly suppressed down to 0.1 and then rapidly enhanced up to 1.1, demonstrating a large modulation depth and a broad dynamic range, due to the progressive transition of dominant modes from the bright state to the low-radiating dark state and further moving towards the bright state again.” (1st paragraph, page 4 in the revised text)

Comment 9) Line 114: ‘The anapole state is an engineered superposition of toroidal and electric multipoles...’ – well, this is not quite exact. Both toroidal and electric dipoles can be non-zero, but anapole state appears when they interfere destructively (also mentioned in line 62). I would rather rewrite this sentence as ‘The anapole state is an engineered destructive interference between toroidal and electric dipoles...’

Reply: We have changed the sentence accordingly into “The anapole state is an engineered destructive interference between toroidal and electric dipoles presenting in well-designed dielectric nanostructures”.

Comment 10) Line 163: ‘Compared with the photothermal induced index change $\Delta n \sim 3 \times 10^{-5}$ in bulk Si under the same laser excitation intensity 41,...’ This is not a fair comparison, because in reference 41 the used wavelength was telecom, at which there is no absorption in Si. What would be nice to compare with is the photothermally induced index change for bulk Si (or for 50-nm-thin Si film used by authors) using the same setup and wavelength.

Reply: To make a fair comparison, we measured the Raman scattering of a standard Si wafer and extracted the temperature elevation under the exposure of the 532 nm

CW laser. Following the same procedure for Si nanodisks, the temperature rises were calculated by measuring the ratio of anti-Stokes to Stokes intensity I_A/I_S in the Raman spectra. The results are added in Supplementary Note 6 and figure S6, as follows:

From Fig. R4, it is clearly seen that bulk single crystalline Si has rather small change of I_A/I_S throughout the range of laser intensities, manifesting that there was less than 10°C temperature rise. This corresponds to $\Delta n \approx 0.005$ when laser intensity I is 12 MW/cm², leading to $n_{2,bulk@532nm} = \Delta n/I \approx 4 \times 10^{-4}$ cm²/MW. Therefore, this result implies a three-order-of-magnitude enhanced photothermal nonlinearity by Si nanodisks supporting anapole states compared with the bulk Si. It is worth noting that in a previous study [ref. 9], it is reported that the photothermal nonlinearity of Si is on the order of $\approx 10^{-5}$ cm²/MW at the telecom-band.

Fig.R4 (a) Raman spectra of a Si wafer taken at four different laser intensities at the wavelength of 532 nm. The Stokes signal is normalized to 1 for better visualization of the ratio of anti-Stokes and Stokes lines. (b) Variation of anti-Stokes-to-Stokes ratio I_A/I_S as a function of irradiance intensities. (c) Extracted temperature rises at different

irradiance intensities.

[9]. Horvath, C., Bachman, D., Indoe, R. & Van, V. Photothermal nonlinearity and optical bistability in a graphene-silicon waveguide resonator. *Opt Lett* **38**, 5036-5039 (2013).

Comment 11) Figure 1b: To support the idea of linear scattering at small intensities, authors could add a fitting straight line (similarly to red dashed line in figures S5 a, d, e, h).

Reply: We have revised the Fig. 1b and added a fitting straight line to indicate the linear scattering at low intensities.

Comment 12) Figure 2e: I am wondering, why for small illumination intensities the normalized cross-section is not 1 exactly? Is it because of large errors for small intensities?

Reply: As pointed out by the reviewer, at low irradiance intensities, a small fluctuation of the intensity can cause variations of the scattering intensity. To take this into consideration and make the presentation clear, we have updated Fig.1b and Fig. 2e by including error bars in the revised manuscript.

Comment 13) Figures 2-4: I am wondering, why blue-white-red colorbar was used in so many images? I mean, usually this colorbar is used for plotting signed values (with zero being white color, positive values as red, and negative as blue). For unsigned maps like Figure 2d a more suitable is a rainbow colormap (like the one in Figure S6), with zero (or min) being darkest and max being brightest colors. It is a matter of taste, so I leave the final choice to authors.

Reply: We thank the reviewer for professional suggestions. In response to this point, we have updated all the figures. For unsigned scattering images including Fig. 2d and Fig. 4, a rainbow colormap was adopted. When describing the field distributions in Fig.3, a hot colorbar was adopted.

Comment 14) Figure 3 and, maybe, other figures: the text is too small to read (especially in panel a).

Reply: We have enlarged the texts and labels for all the figures in the revised manuscript.

Comment 15) Figure 3b: The multipole decomposition results look strange. Was it done for Si disk on a glass substrate or in vacuum? It is because formulas mentioned in corresponding Methods section are valid for even dielectric environment (and it is hard to do it in uneven surrounding). Therefore, other scientists usually do additional simulations for nanoparticles in even surrounding (for example, air), which gives qualitative image of multipoles. Then, when substrate is added, it can introduce small shifts of the resonances, but it won't drastically change the multipole combination. Secondly, scientists usually calculate contributions of multipoles to the total scattering cross-section (the formula can be taken, for example, from <https://doi.org/10.1021/acs.nanolett.7b04200>, see Methods section there). Then multipole contributions and their sum are plotted together with the total scattering cross-section (as example, see Figure S1 in mentioned reference). By comparing the sum of multipole contributions with the total scattering cross-section, one can judge whether enough multipole orders are considered.

Reply: *Firstly*, the multipole decomposition was performed for Si nanodisks in air. Indeed, formulas in Methods are valid for even dielectric environment. When a low refractive-index substrate is added, it can introduce small shifts of the resonances as mentioned by the reviewer, but it won't drastically change the multipole combination (Fig.R5).

Fig. R5 total scattering cross-sections for Si nanodisk ($D=200$ nm, $h=50$ nm in air with and without the glass substrate).

Secondly, the method of electromagnetic multipolar decomposition was benchmarked by Mie theory in which the analytical solution was solved for a spherical silicon nanostructure. It is clearly seen that the scattering cross-section obtained from Mie theory, FDTD calculation and the sum of multipole contributions are in good agreement (Fig. R6).

Fig. R6 Scattering cross-section of a 200-nm sized Si nanosphere in the air.

Finally, we have followed the reviewer's suggestions and revised the results of decomposition of the scattering in Cartesian multipoles (Fig. R7). Here, *Mult sum* denotes the sum of the scattering contributions from all considered multipoles, in which the interference between ED and TD is taken into account. A reasonable agreement between the electromagnetic multipolar decomposition results and the total scattering cross-section confirms that enough multipole orders are considered. In addition, we have updated Fig. 3b in the revised main text and cited the referenced paper in the main text as ref. 31.

Fig.R7 Multipole decomposition of the scattering in Cartesian multipoles for a Si nanodisk (D=200 nm, h = 50 nm).

Comment 16) Authors mention their modification as reversible and repeatable, with Figure 2f as one of the proofs. Another such proof could be dark-field measurements of the spectra before and after the heating (I mean, adding the last to the Figure S1b).

Reply: To confirm the repeatability, we have recorded the dark-field scattering spectra before and after the nonlinear scattering measurements showcasing negligible variations (Fig. R8). The result has been supplemented to Fig. S3b.

Fig.R8 (a) Simulated backward scattering spectra of isolated nanodisks. (b) Dark-field scattering spectrum before and after nonlinear scattering measurements. The dashed lines indicate the excitation wavelength. The inset shows the schematic of a reflective dark-field microscope.

Comment 17) Supplementary Note 1: it would be great to see a final equation for the temperature as a function of anti-Stokes-to-Stokes ratio. Also, the phonon energy $h\Omega$ – was it measured or used as a fitting parameter?

Reply: We have added the final equation as:

$$I_A/I_S = \bar{n}/(\bar{n} + 1) = \exp\left(\frac{-h\Omega}{k_B T}\right)$$

where $h = 6.626 \times 10^{-34} \text{ J} \cdot \text{s}$, $k_B = 1.38 \times 10^{-23} \text{ J/K}$, and $\Omega = c\omega$ with $c = 3 \times 10^{10} \text{ cm/s}$, and ω is the Raman shift in cm^{-1} . In our work, we use the anti-Stokes-to-Stokes ratio at the characteristic Raman shift $\omega = 520 \text{ cm}^{-1}$ for Si to determine the temperature rise. Please see changes in Supplementary Note 4.

Comment 18) Supplementary line 107: n_0 should not depend on T (i.e., ‘ $n(E,T) = n_0(E) + \dots$ ’)

Reply: We thank the reviewer for pointing out the careless mistake. We have corrected it.

Comment 19) Supplementary Figure S4: why sum of CscA and CscB does not equal to the total CscA?

Reply: We thank the reviewer for this professional comment. In fact, the C_{scaF} and C_{scaB} in Fig. S4 were calculated by utilizing two plane monitors close to the Si structure in order to collect most of the forward/backward scattering power. In our simulations, the plane monitor was positioned at ± 400 nm from the simulation center, with a monitor size of 1900 nm by 1900 nm. We fully agree with the reviewer that the plane monitor was not large enough, which yields that the sum of C_{scaF} and C_{scaB} does not equal to the total C_{sca} . It should be noted that the plane monitors were purposely chosen in the calculations of backward scattering C_{scaB} in Figs. 3e-g in the main text to match the maximum collection angle of the objective lens ($NA = 1.4$) used in our experiments. Nevertheless, we have taken the reviewer's suggestion and updated the calculation by using two half-open box monitors which collect all the scattering fields of forward and backward 2π solid angles. This was confirmed by the results shown in Fig.R9 a and b. It is clearly seen that the sum of C_{scaF} and C_{scaB} equals to total C_{sca} using box monitors. Please see updated Supplementary Note 7 and Fig. S7.

Fig.R9 Scattering calculations with plane monitors (a) and half-open box monitors (b).

Comment 20) Supplementary Figure S5: first, the panels are small and a hard to read. Secondly, to simplify the comparison between three disks, it is better to combine plots: S5a, e, and Figure 3c into one plot; S5 c, g, and Figure 3g into another plot, and the same for S5 d, h, and Figure 3h. Finally, why the line and the linear fit in Figure S5d coincides? According to S5c, the scattering cross-section decreases with temperature, therefore it should not be the same straight line as for small intensities.

Reply: We have re-arranged the contents in Fig.S5 to compare the nonlinear photothermal and scattering behaviors between three nanodisks. And we have increased the sizes of each panel for easy reading. Following the reviewer's suggestion to simplify the comparison between the nanodisks, we have unified the coordinates for each comparison such that readers can also clearly see the differences. The revised figure is shown below:

Fig.R10 Simulation of temperature variations and corresponding nonlinear scattering responses of silicon nanodisks with different sizes. The top panels (a)-(c) show the results of nanodisks with $D=200\text{ nm}$ and $h=50\text{ nm}$, same as Fig. 3(c), 3(g), and 3(h) in the main text. The middle panels (d)-(f) show for nanodisks with $D=170\text{ nm}$ and $h=50\text{ nm}$, whilst the bottom panels (g)-(i) show for nanodisks with $D=230\text{ nm}$ and $h=50\text{ nm}$. (a), (d) and (g) are the temperature rises under the given excitation intensity. (b), (e) and (h) are the backward scattering cross-section modulations when increasing the temperature. Note that photothermal tuning near ED mode results in very small change in scattering cross-section (marked by the shadow range in (e)). By combining

the above two sets of figures, (c), (f) and (i) present the resulting simulated nonlinear scattering behaviors. Dash lines denote the linear trends without taking the photothermal nonlinearity into account.

Regarding the reviewer's question about Fig.S5c in previous supplementary information, we would like to clarify this with the revised figure R10 (Supplementary Figure S8-B)). For the Si size of $D=170$ nm, please note that it is ED-mediated photothermal process. Throughout the irradiance intensity range up to 1.5 MW/cm^2 , there is a moderate temperature rise up to 200°C . Meanwhile, the backward scattering cross-section keeps almost unchanged (slightly decreases from about 1.65 to $1.62 \times 10^{-14} \text{ m}^2$, less than 2%, shown in the shadow range) within elevated temperatures. Therefore, the scattering and its linear fit coincide.

Comment 21) Supplementary Figure S6a: I am wondering, why authors decided to fit the dependence with five order polynomial? I don't see it being used anywhere else.

Reply: It is just a polynomial function fitting for best eye guidance. It actually has no physical meanings here. In order to avoid the misleading here, we have changed the legend into "best fit for eye guidance".

Comment 22) Supplementary Figure S6c-f: why all panels in the top show diagonal symmetry? And what was the polarization? Finally, what does these images show?

Reply: We deeply appreciate the careful reading by the reviewer. The elongation is caused by depolarization effects under tight focusing by a high NA objective. The polarization used is linear polarization along the diagonal direction. Without loss of generality, we have re-plotted this figure by using circular polarization which exhibits circular symmetry (Fig. R11). The figure is to demonstrate a set of theoretical simulations nicely reproducing our observations of evolution of PSFs at different excitation intensities in Fig. 2 in the main text given the nonlinear scattering response (shown in Fig.1b in the main text) is known.

Please see revised Supplementary Note 2 and Figure S2 accordingly.

Fig. R11 Point spread function analysis of scattering images. (a) Measured nonlinear scattering dependence on the excitation intensity as well as the fit for eye guidance. (b) Gaussian distribution and the cross-section of the 532 nm excitation laser beam at the focal plane of an objective lens of NA=1.4. (c-f) Calculated nonlinear scattering images and corresponding cross-sections under the different excitation intensities.

Comment 23) Supplementary Figure S7a, right panel: why some holes, according to the colormap, are below 10 nm, when the max is 60 nm (top of disks), and disk height is 50 nm? Is it because glass substrate was also etched between disks?

Reply: The holes shown in the AFM image are defects caused by the vacancy of self-assembled polystyrene spheres during the colloidal lithography, where the glass

substrate is exposed to the etching directly. In addition, it is technically challenging to immediately stop the etching action right after Si thin film is perforated. Therefore, the glass substrate was also etched a little bit between Si nanodisks as pointed out by the reviewer. In order to clarify the height, we have selected two representative regions as shown below. It can be seen that the defect region is etched about 60 nm in depth in the glass substrate, and the heights of the Si disks are $50 \text{ nm} \pm 2 \text{ nm}$. The figure is added as Supplementary Figure S9 in the revised version.

Fig.R12 AFM Characterization of the Si nanodisk array sample. (a) AFM image and zoom-in images (b and c) of Si nanodisk arrays with $D=200 \text{ nm}$ and $h=50 \text{ nm}$, Scale bar: $4 \mu\text{m}$. and (c). Zoom-in images for a region containing uniformly distributed Si nanodisks (b) and a region containing defects (c). The right panels are cross-sections to show the height of the nanodisks.

Comment 24) As I mentioned in the beginning, there are several mistakes and misspellings in the main text. Here are some examples:

- Line 132: ‘Further increases of excitation intensities, a sharp peak emerges...’ – I believe, it should be something like ‘By further increasing the excitation intensity, a sharp peak emerges...’
- Line 305: ‘Linear polarization excitation was obtained by imposing a half-wave plate on the laser beam.’ – Half-wave plate does not give linear polarization; it can only rotate it. I think, authors meant here ‘Linear

polarization excitation was controlled by imposing a half-wave plate on the laser beam.'

c. Line 330: 'Raman spectra were recorded over an acquirement time of 1 s.' I think, usually this is called acquisition time.

d. Line 471: 'Illustration of strong optical heating that efficientLY converts...'

Reply: We thank the reviewer for careful reading. We have corrected these mistakes throughout the manuscript and extensively polished the English in order to improve the readability.

#####

Editorial comments

Comment 1) To improve the quality of methods and statistics reporting in our papers, we are now asking all authors to complete an editorial policy checklist that verifies compliance with all required editorial policies.

Reply: We have completed the editorial policy checklist.

Comment 2) At the same time, we ask that you ensure your manuscript complies with our editorial policies

Reply: We have ensured our manuscript complies with the editorial policies

Comment 3) Data availability statements and data citations policy: All Nature Communications manuscripts must include a section titled "Data Availability" as a separate section after the Methods section but before the References

Reply: The revised manuscript includes the Data availability statement: Data availability. The data that support the findings of this study are available from

the corresponding author upon reasonable request.

Comment 4) Please ensure that all co-authors are aware that they can add their ORCIDs to their accounts and that they must do so prior to acceptance

Reply: All co-authors are aware that they can add their ORCIDs to their accounts and that they must do so prior to acceptance.

REVIEWERS' COMMENTS

Reviewer #1 (Remarks to the Author):

I have carefully studied the revised manuscript by Zhang et al., as well as their replies to all the reviewers' comments. I think the authors did a stellar job on enhancing the main points of the manuscript, which has become much better after revision, and removed all the misleading claims. I suggest accepting this version of the paper for publication in Nature Communications.

Reviewer #2 (Remarks to the Author):

Authors did a tremendous work in revising their manuscript. It looks much better and clearer now. I like that they removed questions about amorphous/crystalline Si and its temperature dependence of n and k . What surprised me is added ellipsometric measurements at elevated temperatures, which improved the solidness of used models and the work itself. I believe this work now fits the quality level of Nature Communications and deserves to be published there after few following minor comments:

- 1) In response to my comment, authors added mentioning of pre-annealing of Si to switch them into crystalline phase. I think it is worth adding similar phrase to the Methods (Preparation of silicon nanodisks), with a reference to Supplementary note 1.
 - 2) Authors did their own measurements of Si temperature dependence of n and k to support used model from the literature. Therefore, I believe, it is worth adding mentioning about it to the Methods (Complex refractive index of Si at elevated temperatures).
 - 3) Please add somewhere the rough estimation of absorbed power per disk, required to raise its temperature to get the refractive index change of 0.5 (to my estimation, it is about 0.1 mW); and estimation of the absorption efficiency, that is, how much power of the incident beam is absorbed by a single disk (to my estimation, about 5%).
 - 4) Line 153: I think it should be 'thermally' instead of 'thermal' in the phrase 'thermal sensitive visible wavelength'
 - 5) Line 167: Similarly, I believe it should be photothermally in the phrase 'photothermal-induced'
 - 6) Line 160-163: The sentence 'Unlike the absorption that is ascribed to all multipole modes of the nanodisk keeps increasing...' is a bit hard to follow; I would recommend rewriting it.
 - 7) Line 265: please write all ' \ln ' as non-italic in the expression of beta (not only the first one). Also, please add the final value of beta for simplicity. Finally, in the same line, for 'Req' eq should be subscript.
 - 8) Line 279: it would sound better to modify the phrase 'And extrapolation was made for determined the complex refractive index...' as 'An extrapolation was made to determine the complex refractive index...'
 - 9) Figure 2f: this figure should confirm the reversibility of nonlinear scattering, which it does only partially: it confirms the reversibility of scattering, but not nonlinear scattering. Nonlinear scattering is observed when the scattering cross-section, rather than the scattering itself, is measured. If authors have the corresponding data, it would be beneficial to plot the evolution of normalized scattering cross-section by switching between suppressed and either linear (at low intensities) or reverse saturation state (RSS).
 - 10) Figure 4: is it a bit misleading, when labels in Figure 4d (SS, RSS-1, RSS-2) are used for different meaning, that is, the latter two correspond to differential images, while the first (SS) is for the image itself. I propose to introduce a new label for differential images, for example DSS or DRSS, and add it to the right panel in (a), both panels in (c), and axis labels in (d). Also, the label confocal is misleading in (b), because all images are confocal; I assume it is better to rename it to linear, since it opposes SS and RSS. Of course, it represents a usual confocal microscope image of the structure, and the figure caption explains it well.
- Supplementary information:
- 11) Line 92: 'objective lens (Olympus, MPlanFLN, 50 \times , NA=0.8,).' – extra comma after 0.8
 - 12) Page 5: please write function 'exp' and units (J/K and others) as non-italic. Also in the rest of

the text, if any

13) Page 6: please write variables (n , k , E , T) in the text as italic. Also in the rest of the text, if any

Good luck!

Vladimir Zenin

Manuscript ID: NCOMMS-20-05933A

Manuscript title: “Anapole Mediated Giant Photothermal Nonlinearity in Nanostructured Silicon”

Point-by-point responses to Reviewers' Comments

We are very grateful for all the comments from the reviewers. These comments are very important and valuable to improve the quality and readability of this paper. Revisions and responses to address your comments are presented as below.

#####

Reviewer #1 (Remarks to the Author):

I have carefully studied the revised manuscript by Zhang et al., as well as their replies to all the reviewers' comments. I think the authors did a stellar job on enhancing the main points of the manuscript, which has become much better after revision, and removed all the misleading claims. I suggest accepting this version of the paper for publication in Nature Communications.

Reply: We appreciate the reviewer's positive comments on our revision and recommendation for publication in Nature Communications.

#####

Reviewer #2:

Authors did a tremendous work in revising their manuscript. It looks much better and clearer now. I like that they removed questions about amorphous/crystalline Si and its temperature dependence of n and k . What surprised me is added ellipsometric measurements at elevated temperatures, which improved the solidness of used models and the work itself. I believe this work now fits the quality level of Nature

Communications and deserves to be published there after few following minor comments:

Reply: We thank the reviewer very much for positive comments and constructive suggestions, which greatly improves the quality of this manuscript to meet the high standards of Nature Communications.

- 1) In response to my comment, authors added mentioning of pre-annealing of Si to switch them into crystalline phase. I think it is worth adding similar phrase to the Methods (Preparation of silicon nanodisks), with a reference to Supplementary note

Reply: Following the reviewer's suggestion, we have added a sentence in Methods (Preparation of silicon nanodisks): "The as-prepared Si samples are pre-annealed to switch into crystalline phase, before performing all the nonlinear scattering measurements (Supplementary Note 1)."

- 2) Authors did their own measurements of Si temperature dependence of n and k to support used model from the literature. Therefore, I believe, it is worth adding mentioning about it to the Methods (Complex refractive index of Si at elevated temperatures).

Reply: We have added a sentence in Methods (Complex refractive index of Si at elevated temperatures): "Measurements of temperature dependence of n and k were performed up to 400°C by ellipsometric techniques, which show good congruence with the model adopted from literature^{39,40}. And then, an extrapolation was made to determine the complex refractive index at high temperatures. "

- 3) Please add somewhere the rough estimation of absorbed power per disk, required to raise its temperature to get the refractive index change of 0.5 (to my estimation, it is about 0.1 mW); and estimation of the absorption efficiency, that is, how much

power of the incident beam is absorbed by a single disk (to my estimation, about 5%).

Reply: We thank the reviewer for the professional suggestions. The absorbed power per disk, required to raise its temperature to get the refractive index change of 0.5 was calculated as $P_{abs} = C_{abs}I$, where C_{abs} is the absorption cross-section and I is the average light intensity on the sample. Considering that C_{abs} is dependent on the temperature and is roughly increased from 1.6×10^{-14} to $3.2 \times 10^{-14} \text{m}^2$ (from RT to 1000°C, also see Fig. 3d for reference) and the light intensity is taken as 1.25MW/cm^2 , the estimated absorbed power per disk is 0.2~0.4mW. Therefore, the estimated absorption efficiency is 9.5~19%.

Following the reviewer's suggestion, we have added this mentioning in the Methods (Microscope system) on page 12: "Under such circumstance, the disk raises its temperature to cause the refractive index change of 0.5. The corresponding absorbed power per disk is estimated to be 0.2~0.4mW, and the estimated absorption efficiency is 9.5~19%."

4) Line 153: I think it should be 'thermally' instead of 'thermal' in the phrase 'thermal sensitive visible wavelength'

Reply: We have corrected this accordingly.

5) Line 167: Similarly, I believe it should be photothermally in the phrase 'photothermal-induced'

Reply: We have corrected this accordingly.

6) Line 160-163: The sentence 'Unlike the absorption that is ascribed to all multipole modes of the nanodisk keeps increasing...' is a bit hard to follow; I would recommend rewriting it.

Reply: The we have rewritten the sentence as: “Since the absorption of the nanodisk depends on the contribution from all multipole modes, it keeps increasing with temperature (Fig. 3d). In contrast, the scattering can be desirably manipulated in response to optical heating of the anapole mode, thus yielding unconventional nonlinear scattering responses.”

7) Line 265: please write all ‘ln’ as non-italic in the expression of beta (not only the first one). Also, please add the final value of beta for simplicity. Finally, in the same line, for ‘Req’ eq should be subscript.

Reply: The types of text font are all carefully checked and corrected, and the final value of beta is added.

8) Line 279: it would sound better to modify the phrase ‘And extrapolation was made for determined the complex refractive index...’ as ‘An extrapolation was made to determine the complex refractive index...’

Reply: The phrase has been modified as suggested by the reviewer.

9) Figure 2f: this figure should confirm the reversibility of nonlinear scattering, which it does only partially: it confirms the reversibility of scattering, but not nonlinear scattering. Nonlinear scattering is observed when the scattering cross-section, rather than the scattering itself, is measured. If authors have the corresponding data, it would be beneficial to plot the evolution of normalized scattering cross-section by switching between suppressed and either linear (at low intensities) or reverse saturation state (RSS).

Reply: We thank the reviewer for the comments. To address this concern, we supplement the figure to confirm the reversibility of the nonlinear scattering process by checking the evolution of normalized scattering cross-section. Please see the

revised Supplementary Figure 3.

Fig.R1 Reversibility of normalized scattering cross-sections by varying the excitation intensities from low to high, and vice versa. The arrows in the figure indicate increasing (black) or decreasing (red) the excitation intensities. The error bars show the standard deviations of normalized scattering cross-sections according to statistics of twelve nanodisks.

10) Figure 4: is it a bit misleading, when labels in Figure 4d (SS, RSS-1, RSS-2) are used for different meaning, that is, the latter two correspond to differential images, while the first (SS) is for the image itself. I propose to introduce a new label for differential images, for example DSS or DRSS, and add it to the right panel in (a), both panels in (c), and axis labels in (d). Also, the label confocal is misleading in (b), because all images are confocal; I assume it is better to rename it to linear, since it opposes SS and RSS. Off course, it represents a usual confocal microscope image of the structure, and the figure caption explains it well.

Reply: We thank the reviewer for very nice suggestions. We have corrected the label in Figure 4d showing different images labeled as Linear, SS, DRSS-1 and DRSS-2, since the first two are the images themselves, and the latter two are the differential images for two RSS images. Also the label “confocal” is substituted with “linear” to

avoid misleading.

Supplementary information:

11) Line 92: 'objective lens (Olympus, MPlanFLN, 50×, NA=0.8,).' – extra comma after 0.8

Reply: We thank the reviewer for the careful reading. The extra comma has been removed.

12) Page 5: please write function 'exp' and units (J/K and others) as non-italic. Also in the rest of the text, if any

Reply: The types of text font are all carefully checked and corrected.

13) Page 6: please write variables (n , k , E , T) in the text as italic. Also in the rest of the text, if any

Reply: All the scalar variables such as (n , k , E , T) are all italicized.